# Mitochondrial Dysfunction as a Pathogenesis and Therapeutic Strategy for Metabolic-Dysfunction-Associated Steatotic Liver Disease

**DOI:** 10.3390/ijms26094256

**Published:** 2025-04-30

**Authors:** Xiangqiong Li, Wenling Chen, Zhuangzhuang Jia, Yahui Xiao, Anhua Shi, Xuan Ma

**Affiliations:** 1School of Basic Medical Sciences, Yunnan University of Chinese Medicine, Kunming 650500, China; sharonli2025@163.com (X.L.); 17330034852@163.com (Y.X.); 18388078016@163.com (X.M.); 2Yunnan Key Laboratory of Integrated Traditional Chinese and Western Medicine for Chronic Disease in Prevention and Treatment, Kunming 650500, China; 3Key Laboratory of Microcosmic Syndrome Differentiation, Education Department of Yunnan, Kunming 650500, China; 4The First Clinical College of Yunnan University of Chinese Medicine, Kunming 650500, China; ynwenling09@126.com

**Keywords:** MASLD, mitochondrion, ROS, autophagy, ATP, NAFLD

## Abstract

Metabolic-dysfunction-associated steatotic liver disease (MASLD) has emerged as a significant public health concern, attributed to its increasing prevalence and correlation with metabolic disorders, including obesity and type 2 diabetes. Recent research has highlighted that mitochondrial dysfunction can result in the accumulation of lipids in non-adipose tissues, as well as increased oxidative stress and inflammation. These factors are crucial in advancing the progression of MASLD. Despite advances in the understanding of MASLD pathophysiology, challenges remain in identifying effective therapeutic strategies targeting mitochondrial dysfunction. This review aims to consolidate current knowledge on how mitochondrial imbalance affects the development and progression of MASLD, while addressing existing research gaps and potential avenues for future research. This review was conducted after a systematic search of comprehensive academic databases such as PubMed, Embase, and Web of Science to gather information on mitochondrial dysfunction as well as mitochondrial-based treatments for MASLD.

## 1. Introduction

Non-alcoholic fatty liver disease (NAFLD) has emerged as a major health issue globally, demanding our attention. In recent decades, its incidence has increased significantly worldwide [1]. According to current accurate estimates, NAFLD affects approximately 25% of the global population [2]. Given that NAFLD can occur concurrently with various other hepatic conditions, such as viral hepatitis and autoimmune disorders, recent discussions among experts have led to a consensus that metabolic-dysfunction-associated fatty liver disease (MAFLD) may serve as a more suitable and comprehensive definition [3]. At EASL 2023, leaders of the multinational liver societies, including the American Association for the Study of Liver Diseases (AASLD) and the European Association for the Study of the Liver (EASL), as well as the co-chairs of the NAFLD Nomenclature Initiative, announced the naming of NAFLD as MASLD, following the international, multisociety-guided Delphi process [4]. Mitochondria, as the intracellular “energy factories”, are inextricably and closely linked to the development of MASLD [5]. Mitochondria play a central role in cellular energy metabolism, generating adenosine triphosphate (ATP) mainly through oxidative phosphorylation, which provides a constant supply of energy for various cellular life activities [6]. The lack of cellular energy supply makes it difficult to maintain normal physiological functions, affecting the liver’s metabolic function. On the one hand, mitochondrial dysfunction leads to excessive accumulation of reactive oxygen species (ROS), which are highly oxidizing and cause severe oxidative damage to intracellular biomolecules such as proteins, lipids, and DNA. On the other hand, metabolites, such as those produced by the tricarboxylic acid (TCA) cycle, are not only key factors in cellular energy metabolism but also have a big say in the regulation of cell growth, differentiation, and apoptosis [7]. As a result, investigating the relationship between mitochondria and MASLD is crucial, as it plays a vital role in uncovering the pathogenesis of MASLD and paving the way for the development of more efficient therapeutic approaches.

## 2. Physiological Functions of Mitochondria

### 2.1. Mitochondrial Structure

Mitochondria are often referred to as the powerhouse of the cell, playing an essential role in energy production and various metabolic processes. In addition, they regulate calcium signaling and its homeostasis, influence and control ROS levels, and stimulate immune responses.

However, through their involvement in cell death, mitochondria can easily shift from regulating the normal cellular state to promoting cellular stress. Structurally, mitochondria are characterized by a two-membrane system consisting of an outer membrane and an inner membrane. The outer membrane is smooth and permeable and contains pores that allow the passage of small molecules and ions, such as voltage-dependent anion-selective channels (VDAC)/porins [8]. The outer membrane has implications for controlling mitochondrial behavior and cellular communication. In contrast, the inner membrane is characterized by higher proteins and forms highly stacked invaginations in the matrix called cristae, which greatly increase the surface area for biochemical reactions. In addition, the permeabilities of the inner and outer membranes differ considerably, with the outer membrane being highly permeable (VDAC). Still, only water, oxygen (O_2_), and carbon dioxide (CO_2_) pass freely through the outer membrane. The variation in selectivity generates an electrochemical gradient across the membrane, which serves as the foundation for adenosine triphosphate (ATP) synthesis [9]. The matrix is a space surrounded by the inner membrane and contains nicotinamide adenine dinucleotide (NADH) and flavin adenine dinucleotide (FADH_2_), both of which are high-energy molecules produced in different metabolic pathways such as glycolysis, fatty acid oxidation, and the citric acid cycle. These molecules carry electrons to the electron transport chain on the inner mitochondrial membrane and participate in the oxidative phosphorylation process to produce ATP. The structural integrity of the mitochondria is critical to their function, as any disruption can lead to impaired energy production and cellular dysfunction [10]. Recent studies have shown that the arrangement and morphology of cristae are critical for efficient ATP synthesis, and this functional compartmentalization not only improves the efficiency of ATP production but also involves mitochondria in the regulation of various metabolic pathways, including reactive oxygen species (ROS) and apoptosis [11]. The quantity of mitochondria exhibits significant variability across different cell types, with estimates indicating approximately 800 mitochondria present in mammalian hepatocytes and up to 100,000 in oocytes.

Oxidative phosphorylation (OXPHOS) is a crucial metabolic pathway within the mitochondria that utilizes enzymes to oxidize nutrients, enabling the liberation of energy vital for cellular respiration and the production of ATP [12].

### 2.2. ATP Production

#### 2.2.1. Structural and Functional Characterization of the Mitochondrial Respiratory Chain

Mitochondria, the “energy factories” within the cell, have a subtle and complex mechanism for producing ATP. Mitochondrial respiration is the process of ATP generation, which occurs at the inner mitochondrial membrane.

Electrons from metabolic substrates pass through an electron transport chain (ETC) consisting of a series of protein complexes [13]. As electrons traverse the ETC, protons (H^+^) are actively transported from the mitochondrial matrix to the intermembrane space, establishing a proton gradient across the inner membrane. When protons flow back into the matrix, ATP synthase rapidly captures the potential energy difference released during this process and, like a waterwheel driven by the difference in water flow, precisely synthesizes adenosine diphosphate (ADP) and inorganic phosphate into ATP, completing the conversion from energy potential to chemical energy. It utilizes the potential energy difference to synthesize ADP and inorganic phosphate into ATP [14]. Specifically, the mitochondrial respiratory chain is located on the inner membrane of the mitochondria and consists of five complexes—complex I (NADH dehydrogenase NADH), complex II (succinate dehydrogenase), complex III (NADPH cytochrome c reductase), complex IV (cytochrome c oxidase), and complex V (ATP synthase)—that work together to perform oxidative phosphorylation, a crucial process for ATP production (Figure 1). ATP is produced through oxidative phosphorylation. During this process, two electrons are shuttled from complex I due to the oxidation of NADH. Moreover, an extra two electrons are transferred from complex II when succinate is oxidized to fumarate. Coenzyme Q10 facilitates the transfer of electrons to complex III. Complex III then transfers these electrons to cytochrome C, which is then attached to complex IV [15]. In this series of transfers, the electrons continue to flow as if along a precisely designed “circuit”, setting the stage for subsequent energy conversion. In conclusion, complex IV facilitates the reduction of molecular oxygen (O_2_) to water (H_2_O), thereby finalizing the electron transfer process within the respiratory chain. It is important to highlight that complex IV facilitates the reduction of molecular oxygen (O_2_) to water (H_2_O). Additionally, complexes I, III, and IV function as proton pumps, enabling the translocation of protons from the mitochondrial matrix to the intermembrane space, thereby operating against the established electrochemical gradient. The quantity of protons translocated differs among the various complexes, with complexes I and III facilitating the passage of four protons each, while complex IV permits the transport of two protons. This specific mechanism of proton transport is instrumental in establishing the proton gradient. Once the intermembrane space is enriched with protons, i.e., the last complex of the chain, ATP synthase allows protons to pass in the direction of the gradient. Because this process is highly efficient, as many as 30–32 ATP molecules can be produced for each molecule of oxidized glucose [16].

#### 2.2.2. Factors Regulating the Efficiency of ATP Production

ATP production by mitochondria is not a fixed attribute; rather, it is subject to modulation by several factors, such as the availability of substrates, the integrity of the electron transport chain, and the morphological characteristics of the mitochondria [19].

For example, it has been shown that the kinetics of mitochondrial ATP production can vary significantly depending on the primary substrate used, highlighting the adaptability of mitochondrial metabolism to different physiological demands. In addition, the interaction between ROS production and ATP synthesis is crucial, as elevated ROS levels can reduce mitochondrial efficiency, especially under stressful conditions such as hypoxia. In conclusion, mitochondria are integral to cellular energy metabolism, and their structural features and dynamic processes underpin efficient ATP production. Comprehending these mechanisms is essential for clarifying the function of mitochondria in both health and disease, given that mitochondrial dysfunction is linked to various pathological conditions, such as metabolic disorders and neurodegenerative diseases [20]. Therefore, delving into the mechanisms of mitochondrial ATP production not only helps us gain a deeper grasp of basic cellular physiological processes but also lays a rock-solid theoretical foundation for devising diagnostic and therapeutic approaches for mitochondrial-related disorders. This is anticipated to yield even greater perks for human health down the road.

## 3. Mitochondria and MASLD

As a central metabolic organ, the liver orchestrates carbohydrate, protein, and lipid homeostasis [21]. Unlike adipose tissue, it does not primarily store lipids but dynamically regulates free fatty acid (FFA) pools through three interdependent pathways: dietary intake, adipose lipolysis, and de novo lipogenesis. Physiological FFA flux involves mitochondrial β-oxidation or cytoplasmic esterification into triglycerides (TGs), which are exported via very-low-density lipoprotein (VLDL) [22]. Chronic lipid overload disrupts mitochondrial bioenergetics, impairing ATP synthesis and ETC function. This metabolic failure, observed in both MASLD patients and animal models, correlates with TG accumulation and compromised cellular homeostasis.

### 3.1. Mitochondria Function as Pivotal Metabolic Hubs Within Hepatic Tissue

#### 3.1.1. Mitochondria and Their Function in Hepatocytes

The liver is responsible for essential metabolic processes and engages in interactions with all bodily tissues and organs. It plays a crucial role in various physiological processes, including digestion, energy metabolism, and detoxification, in addition to its involvement in endocrine functions, blood coagulation, and the synthesis of essential plasma proteins. The liver consists of diverse cell types, like endothelial, epithelial, Kupffer, stellate, and pit cells. Among them, hepatocytes make up around 70–85% of the organ’s mass and are especially prone to cellular damage [23]. In hepatocytes, mitochondria function as crucial organelles that serve as metabolic centers, playing a significant role in the regulation of hepatocyte homeostasis. Mitochondria are highly dynamic organelles that play a central role in energy metabolism. Under typical physiological conditions, the liver utilizes approximately 15% of the total oxygen available to the organism. This high oxygen demand indicates that hepatocytes possess a substantial number of mitochondria. Electron microscopy analyses reveal that each hepatocyte within a liver section contains a considerable quantity of mitochondria (ranging from 500 to 4000), peroxisomes (between 300 and 600), and lysosomes (approximately 270). These organelles are essential for the production of ATP, which necessitates oxygen. Mitochondria make up around 18% of the total volume of hepatocytes and play a crucial role in the liver’s metabolic functions. They help break down nutrients like carbohydrates, lipids, and proteins through oxidation to generate energy [24].

#### 3.1.2. Mitochondrial Dysfunction in MASLD

Mitochondrial morphology and function, encompassing enzyme expression and activity, redox status, and ATP generation, show variation in their compartmentalization within the liver [25]. They play a crucial role in fatty acid oxidation, the Krebs cycle, and ATP synthesis, all of which are vital for sustaining liver function and overall energy metabolism [26]. Studies of mitochondrial cytochrome P4502E1 (*mtCYP2E1*), in wild-type and PPAR α-deficient *mice* fed either a standard diet or a high-fat diet (HFD), have shown that the highest levels of hepatic *mtCYP2E1* have been found in PPAR para-deficient HFD mice, which exhibit the highest NAFLD activity scores (NAS) of all animal groups. MASLD and mitochondrial dysfunction serve a significant function in the pathogenesis of the disease, as impaired mitochondrial function leads to the accumulation of lipids in hepatocytes [27] (Figure 2). 

In the liver tissue of patients with fatty liver disease, mitochondrial dysfunction and oxidative stress have been noted [28]. It is marked by different levels of damage to the mitochondrial ultrastructure, abnormal morphological alterations, a reduction in respiratory chain activity, a depletion of ATP, an increase in the permeability of both the outer and inner membranes, an over-production of ROS, oxidative-stress-induced mitochondrial DNA deletion, and impairment of mitochondrial β-oxidation [29,30,31]. In patients with non-alcoholic steatohepatitis (NASH) and corresponding animal models, there is a notable correlation between elevated levels of the microRNA *miR-21* and increased levels of caspase-2 in hepatic tissues. The activation of *miR-21* via the mTOR/NF-κB signaling pathway leads to the suppression of PPAR-α, thereby worsening mitochondrial dysfunction and causing damage to hepatocytes. In this context, the mitochondrial permeability transition (MPT) pore-opening-related cell death process appears to be a key determinant of hepatocyte death. Mitochondrial dysfunction associated with NASH results in a reduction in cellular ATP levels, potentially leading to endoplasmic reticulum (ER) stress and the subsequent activation of the unfolded protein response (UPR). The UPR is associated with the activation of the ab initio adipogenic pathway and further exacerbates steatosis [32]. The accumulation of lipids in the hepatocyte leads to mitochondrial overload, which results in oxidative phosphorylation. Overload leads to impaired oxidative phosphorylation and increased ROS production. Lipid accumulation in hepatocytes can lead to mitochondrial stress, resulting in disrupted oxidative phosphorylation and increased ROS production [33]. This metabolic dysregulation not only exacerbates hepatic steatosis but also promotes inflammation and fibrosis, which are hallmark features of MASLD [34].
Figure 2(**a**) represents normal mitochondria and liver conditions. Mitochondria in a normal liver have normal mitochondrial kinetics, normal exchange of calcium ions inside and outside the mitochondria, regular mitochondrial autophagy, and normal mitochondrial redox conductance. (**b**) represents abnormal mitochondria and fatty liver condition. Compared to a healthy liver, mitochondria in MASLD are fragmented with excess calcium, reduced chemotactic capacity, and increased ROS production, leading to c-Jun N-Terminal Kinase (JNK) activation [35]. JNK activation itself can induce these same defects in mitochondrial function, constituting a feed-forward loop of mitochondrial dysfunction. Mitochondrial dysfunction in MASLD is also explained by mitochondrial self-defects [36]. Reduced fatty acid oxidation caused by this impairment of mitochondrial function is thought to induce fat accumulation in hepatocytes while impairing insulin signaling.
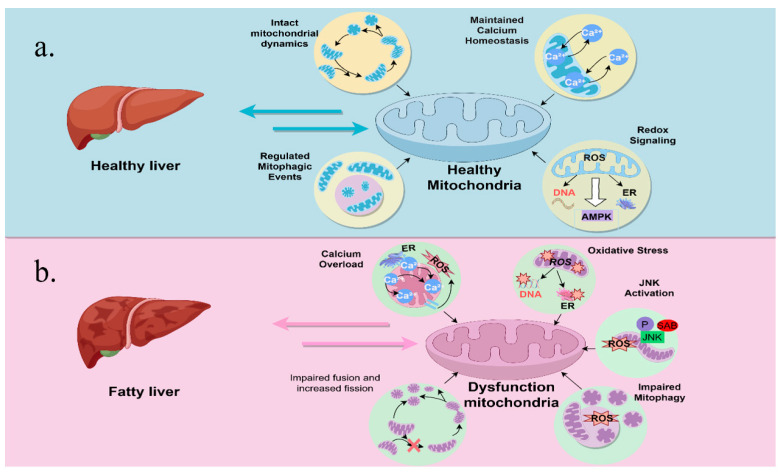


### 3.2. Mitochondrial Adaptation

Mitochondria exhibit significant adaptations to changes in metabolic demands and environmental stresses [37]. Maintaining mitochondrial structure and function is critical for maintaining the complex balance between whole-body dynamic homeostasis and physical contact between organelles (endoplasmic reticulum, lipid droplets, lysosomes, etc.). The safeguarding of cellular physiology necessitates the implementation of various tiers of regulatory mechanisms. The ongoing processes of mitochondrial renewal and degradation play a vital part in controlling the total mitochondrial mass in the cell. In the case of MASLD, mitochondrial adaptive mechanisms (e.g., biogenesis and kinetics) are critical for counteracting the effects of lipid overload and oxidative stress [38].

The mitochondrial unfolded protein response (UPRmt) is governed by signaling pathways that facilitate communication between the mitochondria and the nucleus. This response serves as a compensatory mechanism aimed at restoring and preserving mitochondrial homeostasis in the face of mitochondrial dysfunction or stress, as well as other processes such as mitochondrial autophagy and the dynamics of mitochondrial fission and fusion. In conditions of mitochondrial stress, various transcription factors, including ATFS-1 in the model organism Caenorhabditis elegans and activating transcription factor 4 (ATF4), activating transcription factor 5 (ATF5), and C/EBP homologous protein (CHOP) in mammals, function as mediators of UPRmt. These factors promote the transcription of numerous mitochondrial genes, including those encoding the mitochondrial chaperone mtHSP70, the protease LON, and various antioxidant enzymes. In instances where energy is abundant during the initial adaptation of mitochondria, prolonged overload of FFA can hinder the mitochondria’s ability to oxidize these fatty acids. This impairment subsequently diminishes the translocation of lipids to the mitochondria, mediated by botulinum toxin transaminase-1 (Cpt-1), thereby exacerbating the accumulation of fat within hepatocytes [39]. Simultaneously, impaired β-oxidation results in proton leakage, a reduction in mitochondrial membrane potential, diminished ATP synthesis, and an elevation in ROS production. These changes contribute to insulin sensitivity modifications and the enhancement of antioxidant defenses. At the same time, defective β-oxidation leads to proton leakage, decreased mitochondrial membrane potential, reduced adenosine triphosphate synthesis, and increased ROS production, resulting in altered insulin sensitivity and stronger antioxidant defenses. Understanding these adaptive mechanisms is essential for the development of therapeutic strategies aimed at restoring mitochondrial function in patients with MASLD.

### 3.3. Impaired Mitochondrial Quality Control

Mitochondrial quality control (MQC) is critical for maintaining mitochondrial integrity and function [40]. Mitochondria keep the cellular environment in balance by tweaking their number, shape, and how well they work to meet the increased demands of cellular energy production and stress-handling mechanisms. In the cells of mammals, when mitochondria become damaged, they are singled out and cleared away through a process called autophagy, specifically mitophagy, which is carefully controlled. This process is crucial for preventing the degradation of healthy organelles, a phenomenon referred to as MQC. MQC encompasses several processes, including biogenesis, division, fusion, and mitosis. In the course of these processes, mitochondria initially make every effort to maintain their functionality and structural soundness via means like DNA repair, the workings of antioxidants, protein folding, and degradation routes [41]. Mitochondrial biosynthesis, fusion, and division serve to regulate and maintain mitochondrial function.

Research has shown that autophagy plays a critical role in the elimination of dysfunctional mitochondria. Furthermore, these investigations have indicated the presence of mitochondrial dysfunction and the accumulation of swollen mitochondria in the hepatocytes of *mice* deficient in key autophagy genes, specifically AMP-activated protein kinase (AMPK) and unc-51-like autophagy activating kinase 1 (ULK1) [42]. More recent studies have also observed similar mitochondrial abnormalities in *mice* lacking the Atg7 gene [43]. Research indicates that impaired mitochondria can trigger the opening of the mitochondrial permeability transition pore (MPTP), resulting in membrane depolarization [44]. This process subsequently facilitates the selective phagocytosis of mitochondria by autophagosomes within hepatocytes. The kinetics of fusion and fission processes facilitate the dilution and segregation of impaired mitochondria. In particular, mitochondria exhibiting partial damage undergo fission mediated by dynamin-related protein 1 (DRP1), subsequently leading to their elimination through the process of mitochondrial autophagy [45]. Mitochondria preserve their inherent functionality through the process of fusion, necessitating the implementation of a stringent quality control system. When the initial response proves inadequate, a more comprehensive MQC system is activated.

Impaired mitochondrial autophagy, the selective degradation of damaged mitochondria, can lead to the accumulation of dysfunctional organelles, exacerbate oxidative stress, and promote hepatocyte injury. In addition, failure of mitochondrial quality control mechanisms may lead to the development of more severe forms of MASLD, such as MASH and hepatocellular carcinoma. Therefore, enhanced mitochondrial quality control could serve as a possible therapeutic target for treating MASLD.

### 3.4. Mitochondrial Autophagy Defects

Autophagy is a protective process that provides nourishment during starvation by degrading and recycling cellular components such as excess and/or damaged proteins, macromolecules, organelles, and pathogens by a specialized cellular mechanism, which mediates the phagocytosis of cargoes in membrane structures known as autophagosomes, their fusion with lysosomes, and the subsequent degradation of the cargoes in autophagic lysosomes [46]. Autophagy can be either selective or nonselective. Nonselective autophagy usually occurs during nutrient deprivation, which leads to the extensive breakdown of cytoplasmic components that provide the cell with nutrients for survival. When autophagy is inhibited, it induces cancer and neurodegenerative diseases (e.g., Alzheimer’s disease); when it is over-activated, it may lead to muscular dystrophy. The labeling mechanism of autophagy is typically non-selective, but it can also occur via selective receptors that recognize specific substrates [47]. We provide evidence that mitochondrial autophagy constitutes a selective variant of autophagy, specifically targeting the degradation of impaired mitochondria, which involves the removal of damaged or redundant mitochondrial structures [48] (Figure 3). Mitochondrial autophagy (mitophagy) maintains mitochondrial quality control and prevents ROS accumulation and apoptosis. Mitochondrial autophagy possesses specific recognition of mitochondrial damage markers (e.g., accumulation, BNIP3/NIX expression, etc.). When mitochondrial autophagy is defective, it induces MASLD, Parkinson’s disease, and cardiomyopathy; when it is over-activated, it leads to mitochondrial depletion and triggers an energy crisis [49,50,51]. Mitochondrial autophagy requires disruption of the mitochondrial membrane potential as its effective trigger, and CCCP (a proton-selective ion carrier) and antimycin A (an inhibitor of respiratory complex III) are commonly used to damage mitochondria and activate mitochondrial autophagy [52]. Common autophagy is a fundamental mechanism for cell survival, and mitochondrial autophagy is a “self-regulatory way” to cope with mitochondrial damage.

Autophagy is initiated by the FIP200–ATG13–ULK1 complex, in which the serine/threonine kinase ULK1 plays a central role and is required for the initiation of all types of autophagy including mitochondrial autophagy; it does not directly label the mitochondria but is indirectly involved in mitochondrial autophagy through the activation of downstream effector molecules, such as parkin [53,54,55]. BNIP3 and NIX are related multifunctional mitochondrial outer membrane proteins, with BNIP3 regulating mitochondrial autophagy during hypoxia and NIX being required for mitochondrial autophagy during erythroid lineage development. NIX, BNIP3, and FUN14 domain-containing protein 1 (FUNDC1) all act as receptors that mediate autophagy to scavenge hypoxia-injured mitochondria, which is critically important for reducing ROS levels and maintaining oxygen homeostasis. Second, BNIP3 and NIX compete for binding to B-cell lymphoma 2 (BCL2) (or related proteins), which can release Beclin-1 from the BCL2 complex and activate autophagy [56,57,58]. The most typical mitochondrial autophagy pathway is the PTEN-induced putative kinase 1 (PINK1)/Parkin-dependent pathway. Under healthy conditions, mitochondria have an optimal, relatively high ΔΨm that leads to the degradation of PINK1. However, under unhealthy conditions such as oxidative stress, low ΔΨm leads to the accumulation of PINK1 mitochondria, causing PARKIN to be recruited from the cytoplasm and initiating autophagic degradation of impaired mitochondria, but even in the absence of PINK1, Parkin can be recruited to depolarized mitochondria and drive mitochondrial autophagy [59].

Mitochondrial autophagy plays a crucial role in preserving the integrity of the mitochondrial network, a process that may be compromised with advancing age [60]. Liraglutide, a long-acting glucagon-like peptide-1 (GLP-1) analog, mitigates mitochondrial dysfunction and the generation of ROS while simultaneously promoting PINK1-mediated mitochondrial autophagy and inhibiting lipid accumulation [61,62,63]. Furthermore, the anthocyanin cation cornflowerin-3-O-glucoside has been shown to enhance the condition of MASLD by enhancing PINK1-mediated mitochondrial autophagy in both in vitro and in vivo experimental models. Conversely, liver injury induced by a high-fat diet has been linked to the BNIP3-mediated suppression of mitochondrial autophagy, which is associated with the downregulation of Sirt3 [64,65,66]. This process leads to hepatocytes engaging in a mitochondria-dependent apoptotic pathway. Fatty acids, especially palmitic acid, are capable of firing up hepatic macrophages via the transcription factor hypoxia-inducible factor 1 alpha (HIF-1α). This activation sets off a chain of events that mess with autophagy and pushes macrophages to take on a more inflammatory profile, like ramping up the production of interleukin 1β [67]. Macrophage-stimulated 1 (Mst1) has been identified as a novel upstream regulator of mitochondrial autophagy, which holds considerable importance in modulating apoptosis in cancer cells through the inhibition of mitochondrial autophagy [68]. Mst1 has been identified as a promoter of MASLD through its disruption of Parkin-associated mitochondrial autophagy. Moreover, unsaturated oleic acid (OA) promotes the development of triglyceride-laden lipid droplets, triggers autophagy, and has a negligible effect on apoptosis, which promotes lipid droplet formation poorly and inhibits autophagy but significantly stimulates apoptosis [69].

In addition, mitochondrial damage, oxidative stress, and fatty acid accumulation can be regulated by autophagy. Targeted deletion of ALCAT1, a lysosomal phosphatidyltransferase, in *mice* prevents the onset of NAFLD. alcat1 deficiency also restores mitochondrial autophagy, mitochondrial structure, mitochondrial DNA (mtDNA) fidelity, and oxidative phosphorylation. Mitochondrial autophagy also affects NAFLD. Thyroid hormone (TH) attenuates NAFLD by stimulating mitochondrial autophagy and mitochondrial biogenesis [70], with increased mRNA levels of BNIP3, NIX, ULK1, sequestosome 1(p62), and microtubule-associated protein 1A/1B-light chain 3 (LC3) in response to TH. In addition, the activation of AMPK contributes to a reduction in MASLD through mitochondrial autophagy [71]. It has been shown that reduced mitochondrial autophagy is an early feature of MASLD and accelerates the onset of steatosis and fibrosis. Dysregulation of mitochondrial autophagy may also be influenced by genetic factors. Therefore, targeting the mitochondrial autophagy pathway may show promise for the prevention and treatment of MASLD.
Figure 3The complete process of mitochondrial autophagy in the normal liver. Mitochondrial autophagy receptors or ubiquitin chain/autophagy junctions label mitochondria during mitochondrial stress (loss of membrane potential, mtDNA mutation accumulation, and ROS generation) [72]. These degradation tags recruit the autophagic machinery to the mitochondrial surface and promote the generation of an isolation membrane/phagocytosis that encapsulates the mitochondria [73]. The isolation membrane elongates and closes to form an autophagosome that completely engulfs the target mitochondria. Autophagosomes are transported and fused with lysosomes (vesicles in yeast). Lysosomal hydrolases then degrade the mitochondria.
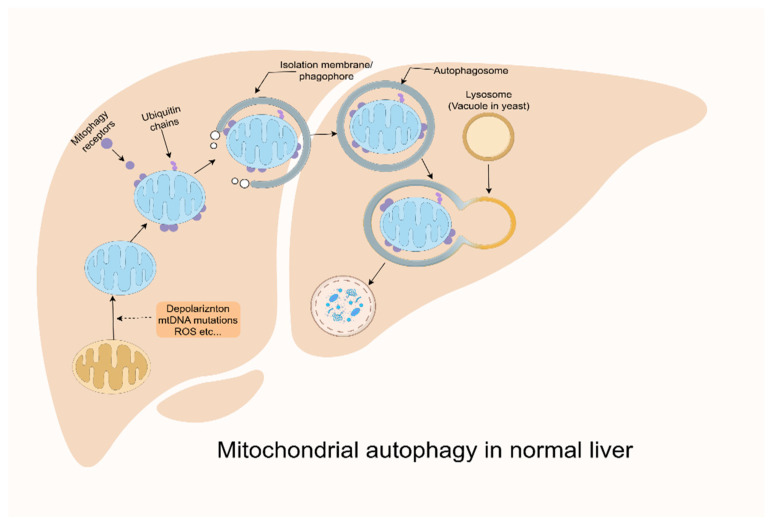


### 3.5. Reactive Oxygen Species and Oxidative Stress

The liver is an important site for ROS production because it is where metabolic and detoxification activities are accomplished [74]. ROS are produced through the mitochondrial respiratory chain ETC and other sources, including peroxisomes, cytochrome P450 oxidase, and NADPH oxidase [75]. These ROS play a crucial role in various signaling pathways and significantly impact numerous biological processes in eukaryotic organisms. However, an imbalance between oxidative and antioxidant can lead to oxidative stress, during which excess ROS cause mitochondrial damage and activate inflammatory and apoptosis-related pathways, leading to cell and tissue damage. In particular, they have the potential to trigger an early detrimental cycle characterized by the autocatalytic inhibition of mitochondrial chain components, which subsequently leads to increased production of ROS.

Several studies have suggested that quercetin may be useful in reducing hepatic lipid accumulation, improving mitochondrial function, and modulating oxidative stress [76]. Quercetin has been shown to inhibit the expression of sterol regulatory element binding protein 1 (SREBP1) and fatty acid synthase (FAS), thereby diminishing lipogenesis and the accumulation of lipids [77]. Additionally, the PI3K/AKT signaling pathway, which plays a crucial role in cell growth and survival, is modulated by quercetin, leading to a reduction in liver inflammation and fibrosis. Furthermore, quercetin influences FXR/TGR signaling, thereby promoting bile acid metabolism and mitigating lipid accumulation. Additionally, quercetin downregulates the TGF-β1/Smads pathway, which is a significant contributor to fibrosis in NAFLD, leading to a reduction in fibrosis markers and potentially decelerating disease progression [78]. Collectively, these pathways underscore the diverse roles of quercetin in tackling the intricate pathophysiology associated with MASLD [79]. Berberine, an alkaloid sourced from different plants in the Berberidaceae family, beefs up the body’s cellular defenses against oxidative stress in multiple ways [80]. It kicks off the AMPK pathway, which, in turn, cuts down the production of mitochondrial ROS and gives a boost to energy metabolism [81].

Antioxidant strategies targeting oxidative stress have shown potential in preclinical studies, emphasizing the importance of mitigating oxidative damage in the treatment of MASLD. For example, curcumin, derived from turmeric, has shown potent antioxidant properties and anti-inflammatory effects. Silymarin, a flavonolignan complex sourced from artichoke seeds, is a herb renowned for its hepatoprotective properties [82]. The antioxidant properties of silymarin are characterized by its ability to inhibit enzymes that generate ROS, stabilize mitochondrial membranes under oxidative stress, and trigger the nuclear factor erythroid 2-related factor 2 (Nrf2) signaling pathway [83]. This pathway stimulates the upregulation of several antioxidant enzymes and helps improve cellular redox balance. Silymarin’s ability to chelate metal ions curbs the formation of highly reactive hydroxyl radicals, which are major factors in oxidative stress [84]. The elevation of systemic oxidative stress and a reduction in serum vitamin E levels observed in patients with NAFLD indicate that the initiation of antioxidant therapy at an early stage may be advantageous, even for individuals diagnosed with uncomplicated fatty liver disease [85]. Vitamin E has demonstrated efficacy in the management of advanced NASH in individuals without diabetes. The PIVENS study, which examined the efficacy of pioglitazone and vitamin E in comparison to a placebo for the treatment of non-diabetic individuals with nonalcoholic steatohepatitis, revealed that the daily administration of 800 IU of vitamin E resulted in significant enhancements in steatosis, inflammation, and hepatocellular ballooning. Furthermore, vitamin E treatment led to the regression of NASH in 36% of the participants, in contrast to 21% in the placebo group [86]. Resveratrol (found in grapes and berries) and green tea extracts have also shown beneficial effects on oxidative stress and liver health in MASLD.

### 3.6. DNA Methylation in Mitochondria

Genetic factors play an important role in MASLD susceptibility, and various studies have identified specific mitochondrial DNA variants associated with the disease [87]. Genetic variations in several genes, including patatin-like phospholipase domain-containing 3 (PNPLA3), transmembrane 6 superfamily member 2 (*TM6SF2*)), membrane-bound O-acyltransferase structural domain 7 (*MBOAT7*), glucokinase regulator (*GCKR*), and hydroxysteroid 17-beta dehydrogenase-13 (*HSD17B13*), have been recognized as significant factors influencing the onset and progression of MASLD [88]. Over the past decade, the corpus of research delving into how aberrant DNA methylation impacts the emergence and development of metabolic disorders, with MASLD being a prime example, has grown by leaps and bounds [89]. The relationships between methylation patterns and transcriptomic profiles were investigated in histologically characterized cohorts of MASLD to determine whether distinctions between mild and advanced MASLD could be identified [90]. The findings indicate that hypomethylation is present in NAFLD across all levels of disease severity when compared to control groups [91]. In addition, the transcription of genes associated with DNA methylation status differed in mild and advanced NAFLD. In advanced MASLD, genes involved in wound healing responses (e.g., fibrogenesis) are hypomethylated, and their expression is upregulated, unlike in the mild form.

The endosymbiotic theory that mitochondria originated from a Gram-negative aerobic bacterium engulfed by a primitive eukaryote that formed a symbiotic relationship with the host cell and evolved into mitochondria is widely accepted. Due to their origin, mitochondria possess their own genome, which is made up of circular double-stranded DNA molecules with a variable number of copies. In humans, mtDNA spans 16.5 kilobases and houses 27 genes. These genes code for 13 proteins of the OXPHOS system. Additionally, there are 22 transfer RNAs (tRNAs) and 2 ribosomal RNAs (rRNAs) involved in the translation of these proteins, and this translation process occurs right inside the mitochondria [92]. For example, re-expression of *Mic19* in *Mic19^LKO^* hepatocytes reverses liver disease in *mice* [93]. The mitochondrial fission protein Drp1 is believed to facilitate the development of MASLD.

Furthermore, the suppression of Drp1 in hepatocytes during the early stages of development has been demonstrated to reduce the occurrence of hepatic steatosis triggered by a high-fat diet in murine models [94]. In patients with NASH, the methylation level of the mitochondrial-genome-encoded NADH dehydrogenase 6 is negatively correlated with the level of physical activity [95]. mtDNA is released from hepatocytes in the context of fatty liver injury, subsequently leading to hepatic inflammation via the activation of toll-like receptor 9 (TLR9) [96]. Diet-induced hepatic steatosis in *mice* involves protein folding and mitochondrial stress response proteins. The mitochondrial chaperone heat shock protein 60 (HSP 60) regulates phosphorylation to counteract high-fat-diet-induced non-alcoholic steatosis [97]. Mt-COX 3 (mitochondria-encoded cytochrome C oxidase III)contains two single nucleotide polymorphism (SNP) sites, specifically nt9821-10A and nt9348A, within the genetic framework of plastic mouse strains. These SNPs are linked to modifications to cellular and mitochondrial adaptation processes as organisms age. Hepatocytes derived from aged mutant *mice* exhibited atypical mitochondrial morphology, elevated levels of mitochondrial ROS, and alterations in mtDNA content [98].

Histological analyses of liver tissue samples revealed significant infiltration and fibrosis. Subsequent investigations indicated the diminished expression of the SOD2 (superoxide dismutase 2, mitochondrial) and FIS1 (fission, mitochondrial 1) genes in aged mutant mice. This observation implies that the reduced lifespan and compromised physical condition of this particular strain may be attributed to mitochondrial dysfunction [99]. Moreover, mtDNA shows a high level of sensitivity to ROS because it lacks histone protection. It is prone to being damaged and mutated under oxidative stress, which results in defects in the respiratory chain and a decrease in mitochondrial biogenesis. Oxidative damage to nuclear DNA adversely affects mitochondrial function and modulates the expression of nuclear-encoded genes associated with mitochondrial activity. For instance, Nrf2, which serves as a critical regulator of antioxidant signaling and plays a significant role in cellular defense against the cytotoxic consequences of oxidative stress, has been documented to exhibit decreased levels in the context of MASLD. When it comes to hepatic metabolism, certain polyunsaturated fatty acids (PUFAs) set off lipid peroxidation. Alongside this, there is an increase in highly reactive aldehyde products like malondialdehyde (MDA) and 4-hydroxy-2-nonenal(4-HNE) [100]. Consequently, these processes might eventually result in mitochondrial harm and a detrimental cycle of mitochondria-generated oxygen radicals. Betaine, a pro-lipotropic substance, can do wonders in reversing the hepatic steatosis brought about by a high-fat diet. It does so by acting on the overall methylation group [101]. Moreover, there is solid evidence backing up the hypomethylation of the CpG cluster in the microsomal MTTP promoter [102]. This promoter plays a key role in the assembly and secretion of VLDL. Understanding these genetic associations could provide an understanding of the pathophysiology of MASLD and help to identify high-risk individuals, facilitating early intervention and personalized treatment strategies.

## 4. Mitochondrial-Based MASLD Treatment

The liver is an important organ for energy metabolism, and energy metabolism dysfunction or metabolic syndrome affects its function, leading to the progression of MASLD and MASH. Therefore, strategies to modulate changes in metabolic dysfunction can be used to treat liver disease [103]. Lifestyle changes are an effective way to prevent and treat MASLD. The primary aim of lifestyle modifications, such as a balanced diet and physical activity, is to maintain a proper weight [104]. Bariatric surgery, also known as weight loss surgery, is regarded as the most bang-for-your-buck treatment for obesity and diabetes. It works its magic by cutting down on food absorption and fine-tuning gut hormone secretion to tackle metabolic dysfunction [105]. In today’s medical field, MASLD has become a highly publicized health challenge worldwide. The incidence of MASLD is increasing significantly owing to changing lifestyles and rising obesity rates. In the process of exploring effective therapeutic strategies for MASLD, antidiabetic drugs, bile acids, mitochondria-targeted drugs, and mitochondrial transplantation have increasingly emerged as key research topics.

### 4.1. Lifestyle Interventions

#### 4.1.1. Dietary Modification

The role of mitochondrial function in the pathogenesis of MASLD is becoming increasingly evident, leading to a growing interest in lifestyle interventions aimed at ameliorating mitochondrial dysfunction. Dietary modifications, such as the adoption of the Mediterranean diet (MD), which is rich in polyunsaturated fatty acids [106], have been shown to enhance mitochondrial biogenesis and function, leading to improved metabolic health. The Mediterranean diet is characterized by a high proportion of monounsaturated fatty acids (MUFAs) and saturated fatty acids (SFAs), with total fat accounting for 30–40% of daily energy consumption, and is based on fruits and vegetables, fish, whole grains, legumes, and olive oil [107]. The EASL-EASD-EASO clinical practice guidelines recommend the MD as an optional diet for the treatment of MASLD. The MD is considered an optional diet for the treatment of MASLD because it improves metabolism by lowering IR and lipid concentrations, induces the regression of steatosis, and significantly mitigates cardiovascular events [108]. A meta-analysis by Takumi Kawaguchi et al. also confirmed that MD improves hepatic steatosis and IR in NAFLD patients. More recently, clinical trials have provided evidence of the feasibility of MD in adolescents and children with NAFLD [109]. A low-calorie ketogenic diet is associated with increased levels of lipocalin, which demonstrates anti-inflammatory properties and influences insulin sensitivity by promoting glucose utilization and fatty acid oxidation [110]. Notably, the therapeutic regimen exhibited dual metabolic and anti-inflammatory effects, manifesting as decreased circulating TNF-α levels alongside improved glycemic regulation, evidenced by HbA1c reduction. Concurrent lipid profile optimization was demonstrated through statistically significant lowering of triglyceride, total cholesterol, and LDL particle concentrations. Conversely, it is associated with elevated levels of high-density lipoproteins (HDLs) and interleukin-10 (IL-10), an inflammatory mediator [111]. Furthermore, research indicates that lipocalin may enhance mitochondrial oxidative stress within hepatic tissue. Ketogenic diet (KD) consumption significantly alters mitochondrial flux and hepatic redox status and promotes ketone body production without affecting intrahepatic triglyceride synthesis, thereby significantly increasing visceral fat content [112]. Time-restricted feeding (TRF) constitutes a circadian rhythm synchronized diet in which access to food is restricted to a specific time window (8–16 h) [113]. A recent study in patients with NAFLD showed that TRF contributed to significant weight loss and lower triglyceride levels after 12 weeks compared to controls [114]. Interestingly, TRF also attenuated the effects of maternal HFD feeding on fetal hepatic steatosis. Other dietary strategies, such as the KD and MD, have also been reported to have epigenetic effects [115]. Astaxanthin (ASTX) is a lutein-like carotenoid found in marine animals such as salmon and shrimp. Studies have shown that ASTX has antioxidant, anti-inflammatory, antidiabetic, and anticardiovascular effects [116]. In C57BL/6J mice with high-fat-diet-induced hepatic steatosis, ASTX depletion (6 or 30 mg/kg of body weight) over an 8-week period, upregulated fatty acid oxidative genes, such as carnitine palmitoyltransferase-1 alpha (Cpt1α) and acyl-coenzyme A oxidase 1 (Acox1), and enhanced the protein expression of Ppara, leading to reduced hepatic lipid accumulation [117]. In addition, ASTX stimulated the expression of thermogenic genes, such as uncoupling protein 2 (Ucp2), in the livers of diet-induced obese (DIO) mice. The study suggests that ASTX is a promising prophylactic and therapeutic agent for the treatment of liver diseases associated with mitochondrial dysfunction [118].

#### 4.1.2. Physical Activity

Regular exercise improves mitochondrial oxidative capacity and promotes fatty acid oxidation, which may significantly reduce hepatic steatosis. Engagement in physical activity and endurance training has been shown to enhance mitochondrial biogenesis and autophagy while simultaneously decreasing the opening of the mitochondrial permeability transition pore (mPTP) in rats subjected to an HFD [119]. Regular physical activity has been correlated with a rise in the number of straight mitochondria in the liver, epigenetic modifications of mitochondrial DNA (specifically, hypermethylation of the MT-ND6 gene), and heightened severity of MASLD [120]. Furthermore, physical activity has been shown to improve MASLD through several mechanisms, including a reduction in intrahepatic fat accumulation, enhancement of fatty acid β-oxidation, stimulation of hepatoprotective autophagy, upregulation of peroxisome proliferator-activated receptor γ (PPAR-γ), mitigation of apoptosis, and improvement of insulin sensitivity [121]. It is not just in the liver that physical activity can bring back mitochondrial function. Studies have shown that exercise can set correct mitochondrial oxygen consumption in ovariectomized rats. Even after the ovaries are taken out, engaging in some physical exercise can offset the harm inflicted by ovariectomy by stimulating mitochondrial function. Exercise has been found to kick-start pathways like SIRT1, which plays a role in mitochondrial dynamics and helps keep energy balance in check in hepatocytes [122]. The combination of resistance and aerobic exercise has been shown to be more rational and effective in clinical practice. For example, resistance exercise is relatively safe and effective in improving the metabolic status of patients with MASLD. A 12-week course of resistance exercise, including push-ups and squats, may help prevent the progression of MASLD [123]. Aerobic exercise showed similar effects. Improvement in histological endpoints of MASLD following a 12-week aerobic exercise intervention improved liver fibrosis. In addition, 12 weeks of high-intensity interval training (HIIT) exercise reduced blood glucose levels and waist circumference in patients with NAFLD [124]. Both HII and 8 weeks of moderate-intensity continuous (MIC) aerobic exercise can reduce intrahepatic triglyceride (IHTG) and visceral fat in obese type 2 diabetes mellitus (T2DM) NAFLD patients. Strength training exercises, which improve muscle strength and increase metabolism, are particularly beneficial for people with NAFLD because they can help control weight and improve insulin sensitivity. In addition, strength training can help prevent muscle mass loss, which is important for overall health and mobility [125].

#### 4.1.3. Weight Loss

Weight loss, particularly through calorie restriction or bariatric surgery, has been associated with improved mitochondrial function and reduced hepatic inflammation in patients with MASLD [126]. The AALSD Practice Guidelines recommend that foregut bariatric surgery be considered for obese patients with MASLD or NASH [127]. Roux-en-Y gastric bypass ameliorated hepatic steatosis in diet-induced obese mice, which was mediated by a mechanistic targeting of the mTOR/AKT2/insulin-induced gene 2 signaling pathway [128]. After Roux-en-Y gastric bypass, NAFLD is attenuated by weight loss and a reduction in steatosis [129]. Laparoscopic sleeve gastrectomy and Roux-en-Y gastric bypass are the most common bariatric surgical procedures designed to reduce the size of the stomach. The mechanism by which they reduce NAFLD appears to be a reduction in oxidative stress and inflammation. At 1 year after laparoscopic sleeve gastrectomy, patients with NAFLD showed significant improvement in liver histology, oxidative stress, and inflammation [130]. Laparoscopic sleeve gastrectomy led to improved histology and subsequent remodeling of cellular interactions, resulting in reduced liver injury in MASLD [131] in a controlled intervention study on bariatric surgery. Several trials have shown that biliopancreatic diversion reverses systemic insulin resistance and reduces inflammation, thereby alleviating NASH [132]. In addition, duodenojejunal bypass, a key component of bariatric surgery, alters lipid metabolism, inflammatory response, and insulin sensitivity in diet-induced obese rats with NASH [133]. Research also suggests that the levels of proteins associated with new lipogenesis in the liver are suppressed after physical activity [134]. Sure enough, this alteration bears a resemblance to multiple markers of the mitochondrial OXPHOS machinery. For instance, there is an upsurge in citrate synthase, palmitate oxidation, and β-hydroxyacyl coenzyme A dehydroacenaphthene as well as in the activity of palmitoyl coenzyme A transferase 1, cytochrome c, and ETC complex IV. Other effects include an increase in the phosphorylated form of acetyl coenzyme A carboxylase (ACC) and a decrease in the activity of enzymes involved in de novo hepatic lipogenesis, including ACC, FA synthase, and stearoyl coenzyme A desaturase (SCD) [135]. Studies have shown that even modest weight loss can lead to substantial improvements in liver histology and function, highlighting the importance of lifestyle changes in the management of MASLD [136]. Integrating these general therapeutic approaches can potentially restore mitochondrial health and attenuate the progression of MASLD, emphasizing the need for a comprehensive approach that includes dietary, physical, and behavioral changes [137].

### 4.2. Pharmacologic and Other Therapies

#### 4.2.1. Antidiabetics

Pharmacological interventions targeting mitochondrial function have emerged as promising strategies for the treatment of MASLD. Antidiabetic medications like metformin enhance mitochondrial performance and boost insulin sensitivity, potentially aiding their effectiveness in treating MASLD [138]. CypD-KO mice treated with metformin for 4 weeks showed insulin sensitivity hepatic neovascularization improvement and, more importantly, elevated levels of mitochondria-associated ER membranes (MAMs) in the liver. Thus, metformin may improve islet cell sensitivity by restoring MAM levels in the liver of mice. Metformin has mixed results; for instance, a meta-analysis that included 14 RCTs found that metformin improved insulin resistance and liver enzymes (ALT/AST) but had no significant effect on liver fat content and histologic scores (e.g., fibrosis), suggesting a limited role in the progression of NAFLD pathology [139]. At the shorter duration of metformin treatment (6.57 weeks), the Emax for NAFLD was shown to be −4.7%, which does not clearly indicate a mixed treatment effect, but may simply indicate a difference in effect [140].

Pioglitazone is an antidiabetic drug belonging to the thiazolidinediones (TZDs) class of drugs, which increase insulin sensitivity by increasing glucose uptake (in, e.g., fat, skeletal muscle, and the liver) and decreasing glucose production by the liver [141]. Cases of liver failure are rare with currently approved TZDs, but they are usually considered safe by routine liver function monitoring in the setting of abnormal liver function. Consensus on the use of TZD therapy in biopsy-proven NASH appears to be limited to specific T2DM patients for whom such therapy is indicated and approved [142]. Pioglitazone has also shown promise in the treatment of steatosis, fibrosis, and inflammation in MASLD and NASH. In the first placebo-controlled human trial, 6 months of pioglitazone treatment reduced hepatic steatosis, hepatocellular ballooning, and inflammation [143]. In a diet-induced model of crab-eating monkeys, both DSCR-induced weight loss and pioglitazone treatment improved the histologic features of NASH after 24 weeks in 30% of the pioglitazone group compared with 0% of the carrier group [144]. Metformin is an AMPK activator, and pioglitazone mainly targets PPARγ, which regulates the transcription of a large number of genes related to insulin and lipid metabolism. Therefore, metformin and pioglitazone have a synergistic effect on the reduction in FFA and triacylglycerol (TAG) synthesis [145].

#### 4.2.2. Bile Acids

The mechanisms based on the gut–liver axis that trigger MASLD include the following: endotoxemia, bile acid metabolism, and endogenous ethanol. Bile acids are key messengers between the host and the gut microbiota, and they play an important role in the regulation of MASH through the activation of various receptors, including the farnesoid X receptor (FXR) and Takeda G protein-coupled receptor 5 (TGR5) [146,147,148]. In gastrointestinal and liver cancer, gut symbionts alleviate MASH through a secondary bile acid biosynthetic pathway. Metabolomics and lipidomics studies in MASLD have identified biomarkers and non-invasive diagnostic tests. FXR is endogenously activated by bile acids and is highly expressed in both hepatocytes and intestinal epithelial cells, where it controls all aspects of bile acid metabolism, including synthesis, export to the bile ducts, and uptake from the intestine. Here, we demonstrate that the inhibition of mitochondrial endomembrane fusion in hepatocytes due to optic atrophy 1 (OPA1) deficiency prevents the development of hepatic steatosis and metabolic syndrome in HFD-fed mice as a result of reduced lipid uptake due to decreased secretion of BAs. At the molecular level, OPA1 deficiency alters the mitochondria–peroxisome–endoplasmic reticulum (ER) axis in the liver, thereby affecting the bile acid (BA) pathway and resulting in the retention of unconjugated BAs and reduced intestinal release of conjugated BAs [149]. In the realm of bile acids (BAs), the naturally occurring “tertiary” dihydroxy ursodeoxycholic acid (UDCA), which is distinct from goose deoxycholic acid (CDCA), has demonstrated a multitude of hepatoprotective effects. It has also enhanced the liver condition of people suffering from various chronic liver diseases. UDCA is an endogenous synthetic bile acid in the body that has antioxidant and anti-inflammatory properties and prevents mitochondrial dysfunction in the progression of obesity-related diseases. A randomized controlled trial by Ratziu et al. demonstrated that treatment with high-dose (28–35 mg/kg/d) UDCA for 12 months lowered hepatic aminotransferase levels and improved glycemic control and IR in patients with NASH. In addition, UDCA has a favorable safety profile, with no deterioration in liver function, and the main adverse effects were abdominal discomfort and diarrhea [150]. At the mitochondrial scale, lipophilic BAs, such as deoxycholic acid (DCA), CDCA, and lithocholic acid (LCA), put the brakes on the ETC. High concentrations of BA (100 μmol/L) had a rather scattergun effect on the intact inner mitochondrial membrane (IMM). On the flip side, low concentrations of BA (10 μmol/L) had a more targeted impact, whether on broken or intact mitochondria, specifically impairing complexes I and III. At the cholestatic level, the effects of UDCA, which is external to CDCA, and LCA inhibit ETC. Recent studies have shown that BA reduces VLDL secretion and serum TG and counteracts hepatic steatosis in a hypertriglyceridemic mouse model [151]. Subsequent research has indeed backed up these effects. In these studies, FXR agonists were discovered to cut down on circulating TG and steatosis. The FXR–SHP axis coordinates this advantageous reshaping of lipid metabolism. This axis puts the brakes on sterol regulatory binding protein-1c (SREBP-1c), a key player in governing hepatic de novo lipogenesis [152], as well as FXR-dependent disruption of ChREBP binding to the liver pyruvate kinase (LPK) promoter, thereby orchestrating this beneficial remodeling of lipid metabolism. Currently, there are limited therapeutic options available for patients diagnosed with NALFD/NASH, but agonists of the nuclear receptor FXR have shown considerable promise. FXR agonists, including obeticholic acid (OCA), a synthetic bile acid coupler, as well as nonsteroidal agonists, are currently being tested in clinical trials as potential therapeutic options for NAFLD/NASH (FXR activation protects against NAFLD via bile-acid-dependent reductions in lipid absorption and obeticholic acid for the treatment of non-alcoholic acid) [153].

#### 4.2.3. Mitochondrial-Targeting Agents

Mitochondria-targeting agents are a class of drug delivery systems that specifically deliver drugs, antioxidants, or sensor molecules, among others, to mitochondria. Resveratrol, a powerful activator and polyphenol present in red wine, can curtail lipid build-up and trigger fatty acid β-oxidation in palmitate-treated hepatocytes in an in vitro setting. Additionally, it increased autophagy and SIRT1 activity. SIRT1 activates PGC-1α, leading to its translocation to the nucleus, where deacetylated PGC-1α enhances the transcriptional activity of nuclear respiratory factor 1/2 (NRF-1 and -2). NRF-1 and -2 then bind to the promoters of response genes involved in mitochondrial biogenesis, energy production, and OXPHOS [154]. In contrast to SIRT1, SIRT3 not only directly activates key enzymes—such as succinate dehydrogenase (SDH), long-chain acyl-CoA dehydrogenase for fatty acid oxidation, and OXPHOS complexes I-IV, including NADH dehydrogenase, SDH, ubiquinol–cytochrome c oxidoreductase, cytochrome c oxidase, and ATP synthase membrane subunit c motif 1in the TCA cycle—but also indirectly activates PGC-1α and AMPK [155]. Moreover, the activation of the Wnt/β-catenin signaling pathway in vitro spurred the differentiation of HPCs into mature hepatocytes. All in all, this led to the mitigation of HFD-induced MASLD in MASLD *mice* [156]. In addition, resveratrol ameliorated hepatic lipid accumulation in KKAY *mice* with intrauterine growth retardation and HFD-fed *mice* by inhibiting mitochondrial dysfunction, oxidative stress, and inflammation. Resveratrol supplementation at 400 mg/kg/day for 15 weeks in C57BL/6J *mice* fed a high-fat diet increased energy expenditure by increasing mitochondrial size and mtDNA content in brown adipose tissue (BAT) through the activation of SIRT1 and PGC-1α. A total of 108 Zucker diabetic rats supplemented with resveratrol (200 mg/kg body weight) in the diet for 6 weeks showed an increase in uncoupled mitochondrial respiration, complex I- and II-supported respiration, and mitochondrial content in the WAT, leading to an increase in glycerol isomerization and lipocalin secretion [157,158,159].

Pterostilbene plays a key role as a free radical scavenger that prevents DNA damage induced by oxidative stress [160]. Pterostilbene acts as an anti-inflammatory molecule as it reduces the number of rats exhibiting moderate inflammation at a dose of 15 mg/kg body weight/d. In addition, no rats showed moderate inflammation, and two rats did not show any inflammation at all after 30 mg/kg bw/d treatment. The molecule partially prevented the increase in hepatic Il-1ß gene expression induced by high-fat, high-fructose feeding (−18% in the PT15 group and −48% in the PT30 group), and *TNFα* expression was significantly reduced. Tanacetin also reduced steatosis, with lower NAS score values in the PT group (3.8 ± 0.3 and 3.1 ± 0.3 for PT15 and PT30, respectively) than in the HFHF group (5.4 ± 0.4) [161]. Pretreatment with pterostilbene was highly effective in curbing lipid build-up in HepG2 cells treated with OA and PA, as well as in Telosapo mice. This was accompanied by an upsurge in the levels of Nrf2, PPAR-α, and heme oxygenase-1 (HO-1), a downturn in the mechanistic target of rapamycin complex 1 (mTORC1) and SREBP-1c, and the kick-starting of both AMPK and autophagy processes. The CP activator cyanidin-3-glucoside (C3G) regulates the brown coloration of uncoupling protein 1 (UCP1) transcripts [162]. Adipose tissue, particularly brown adipose tissue (BAT), shows increased UCP1 transcript activity (UCP1 is expressed in adipose tissue), with higher mitochondrial numbers and upregulation of genes associated with mitochondrial biogenesis in BAT and subcutaneous WAT. This activity facilitates the safe disposal of FFAs and prevents the accumulation of non-esterified FFAs and their destructive effects on mitochondria, slowing the progression of NAFLD. Liver-targeted mitochondrial uncoupling enhances hepatic fat oxidation, leading to a reduction in TAG, plasma membrane (PM) sn-1,2-DAG levels, and protein kinase C epsilon (PKCε) translocations. This process improves hepatic insulin sensitivity, showing therapeutic potential for MASLD, MASH, and T2D [163].

#### 4.2.4. Mitochondrial Transplantation

Mitochondrial transplantation is an innovative area of research aimed at the direct restoration of mitochondrial function in hepatocytes. While mitochondrial transplantation is currently in its early stages, initial research has indicated that this approach may alleviate mitochondrial dysfunction and enhance liver function in models of MASLD [164]. These pharmacologic and advanced therapeutic strategies have great potential for MASLD treatment as our understanding of mitochondrial biology deepens. With the discovery of several limitations of mitochondrial transplantation in scientific experiments and clinical applications, the use of many studies in clinical research involving humans emphasizes the need for rapid isolation of mitochondria under cryogenic conditions, as they are fragile and quickly lose their viability. In addition, appropriate delivery mechanisms must be selected that are consistent with the therapeutic approach in order to accurately target specific tissues [165]. Although autologous mitochondrial transplantation can attenuate immune rejection and inflammation, it is not feasible in patients with systemic mitochondrial defects (e.g., patients with congenital mitochondrial disease), thus requiring the use of mitochondria from individuals of the same species (allogeneic transplantation). Homologous mitochondria may cause marked inflammation that appears to be associated with dysfunction of circulating mitochondrial DAMP-derived homografts [166] (Table 1). In addition, mitochondrial transplantation techniques may raise transgenerational genetic risks (e.g., nuclear–mitochondrial genome mismatch), which is an ethical issue that we cannot afford to ignore.
ijms-26-04256-t001_Table 1Table 1Treatment of MASLD.CategoriesMechanism of ActionClinical EffectivenessMediterranean diet [105,167,168]Regulates lipid metabolism; reduces inflammation; improves insulin resistance.Enhances mitochondrial biogenesis and function, resulting in improved metabolic health and reduced liver fat accumulation.Low-calorie ketogenic diet [110]Reduces liver fat synthesis; promotes fatty acid oxidation; reduces inflammatory response and oxidative stress.Increases lipocalin; decreases levels of TNF-α, glycosylated hemoglobin (HbA1c), and lipids; increases levels of HDL and the inflammatory mediator IL-10.Physical exercise [122]Promotes fat oxidation and decomposition; improves insulin receptor function; improves insulin signaling pathway.Mitochondrial oxidative capacity promotes fatty acid oxidation, mitochondrial biogenesis, and autophagy and reduces HDL.Lose weight [137]Enhances lipolysis and transport; increases insulin sensitivity; regulates blood glucose and lipid metabolism.Mitochondrial function in patients with MASLD is associated with reduced hepatic inflammation and suppression of protein levels associated with hepatic neolipogenesis.Antidiabetic drugs [138]AMP-activated protein kinase; AMPK-dependent changes in cellular energy charge.There is little beneficial effect on hepatic steatosis and inflammation and no effect on hepatic fibrosis and MASH regression.Farnesol X receptor [152]Promotes bile-acid-mediated lipid excretion; regulates glucose metabolism-related genes.Reduces VLDL secretion and serum TG and counteracts hepatic steatosis.Astragalus [162]Mitochondria-targeted agonist.Upregulation of Nrf2, PPAR-α, and HO-1; downregulation of mTORC1 and SREBP-1c; activation of AMPK and autophagy.Mitochondrial transplantation [164,169]Mitochondria-targeted agonist.Direct restoration of mitochondrial function in hepatocytes.

## 5. Conclusions

Mitochondrial dysfunction plays a key role in comprehensively analyzing the pathogenesis of MASLD. As the core of cellular energy metabolism and a hub for the regulation of many physiological functions, mitochondrial dysfunction has a profound impact on the multiple metabolic pathways involved in the process of metabolism-associated fatty liver disease. From lipid metabolism imbalance leading to excessive fat deposition in the liver to energy metabolism disorders leading to insufficient cellular energy supply, which in turn interferes with the normal physiological functions of the liver, mitochondrial dysfunction is present in all aspects of the pathogenesis of MASLD and has emerged as an important factor in the progression of the disease. This large body of in-depth evidence clearly and strongly emphasizes the need for a multipronged approach to treating MASLD. At the same time, the restoration of mitochondrial function is an essential strategy. Repairing and strengthening mitochondrial function can correct metabolic disorders, reduce oxidative stress damage, and radically alleviate the pathological changes of MASLD. It is believed that through in-depth research on mitochondrial function and its relevance to MASLD we can lay a solid foundation for tFhe development of innovative therapeutic programs and bring more hope to patients with MASLD, thus better addressing this increasingly prevalent and serious threat to human health.

## Figures and Tables

**Figure 1 ijms-26-04256-f001:**
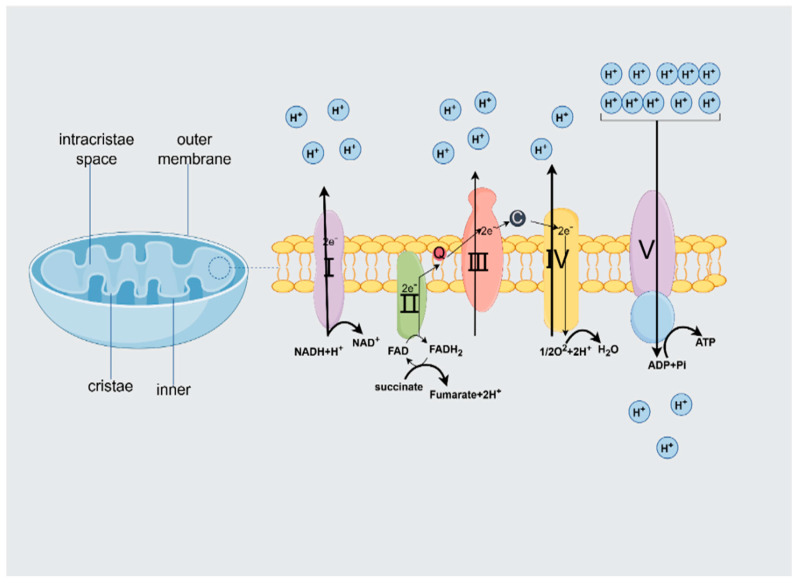
Transmission processes in the mitochondrial electron respiration chain. Mitochondria consist of a double membrane—an inner membrane and an outer membrane—which forms an invagination called a ridge where the OXPHOS complex is located [17]. The mitochondrial matrix contains multiple copies of mitochondrial circular DNA and ribosomes. The mitochondrial respiratory chain is located in the inner membrane of the mitochondria and consists of four complexes and two coenzymes that allow the production of ATP by oxidative phosphorylation [18].

## Data Availability

Not applicable.

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
