# Peer review of "Mitochondrial Dysfunction as a Pathogenesis and Therapeutic Strategy for Metabolic-Dysfunction-Associated Steatotic Liver Disease"

_ijms, 2025, doi:10.3390/ijms26094256_

Round 1

Reviewer 1 Report

Comments and Suggestions for Authors

Thank you for the opportunity to review this interesting and informative manuscript, “Mitochondrial dysfunction as a pathogenesis and therapeutic strategy for nonalcoholic fatty liver disease.” Below, please find my comments:

  • The introductory sections (sections 1 and 2) are somewhat repetitive in places: you introduce NAFLD’s global prevalence and rename it MAFLD in two different segments. Consolidating these details would sharpen the narrative.
  • You correctly mention the shift from “NAFLD” to “MAFLD,” explaining that the term “MAFLD” underscores metabolic dysfunction. Ensure the manuscript consistently uses “MAFLD” once you have introduced it, unless describing older data specifically labeled “NAFLD.” Consider adding a brief discussion on the consensus statements or guidelines (e.g., EASL or AASLD) to give readers context about current nomenclature usage.
  • In some areas (e.g., section 3.4 on mitochondrial autophagy defects), please clarify the distinctions between general autophagy and selective mitophagy. For instance, you mention ULK1, BNIP3, Pink1/Parkin—expanding just slightly on how these pathways interact or differ could help readers unfamiliar with the nuances of mitophagy.
  • You discuss lifestyle and pharmacological interventions well. It would be helpful to briefly address the evidence level for each approach. For instance, metformin has shown mixed results in NAFLD, but pioglitazone has been more studied for NASH. Adding a concise comment on whether each approach has robust clinical trial data, preliminary animal data, or is primarily theoretical would give the reader more perspective.
  • The mention of mitochondrial transplantation is intriguing. As this is a very new approach, you might add a short note on its current limitations (e.g., feasibility, safety in humans) to temper expectations.

Author Response

Comments 1: The introductory sections (sections 1 and 2) are somewhat repetitive in places: you introduce NAFLD’s global prevalence and rename it MAFLD in two different segments. Consolidating these details would sharpen the narrative.

Response 1: Thank you very much for professional comments and suggestions for our work.We have carefully considered your suggestions, and have revised our manuscript accordingly.During the revision process, we systematically reviewed the contents of sections 1 and 2 of the Introduction, deleted repetitions of the global incidence of NAFLD and the introduction of the nomenclature MAFLD, and optimized the paragraph structure.We thank the reviewers again for this valuable suggestion.

Comments 2: You correctly mention the shift from “NAFLD” to “MAFLD,” explaining that the term “MAFLD” underscores metabolic dysfunction. Ensure the manuscript consistently uses “MAFLD” once you have introduced it, unless describing older data specifically labeled “NAFLD.” Consider adding a brief discussion on the consensus statements or guidelines (e.g., EASL or AASLD) to give readers context about current nomenclature usage.

Response 2:We deeply appreciate your professional insights and recommendations regarding our work. Your valuable suggestions have been meticulously deliberated, and we have made the corresponding revisions to our manuscript. I have accurately expressed the definitions of the specialty disease names “NAFLD,” “MAFLD,” and “MASLD” in the introductory section and added EASL and AASLD guideline citations.We would like to reiterate our appreciation to the reviewers for their insightful and constructive suggestions.

 Comments 3: :In some areas (e.g., section 3.4 on mitochondrial autophagy defects), please clarify the distinctions between general autophagy and selective mitophagy. For instance, you mention ULK1, BNIP3, Pink1/Parkin—expanding just slightly on how these pathways interact or differ could help readers unfamiliar with the nuances of mitophagy.

Response 3: We are very grateful for the reviewers' comments, which have important guiding significance for our manuscript.I have already clarified the difference between autophagy in general and mitochondrial autophagy in some respects (Section 3.4 on mitochondrial autophagy defects) by searching scholarly databases, such as PUBMED, in the language of specialized academic terms, and have also added the interactions of the pathways ULK1, BNIP3, and PINK1/Parkin.Once more, we extend our sincere gratitude to the reviewers for their invaluable suggestions.

Comments 4: :You discuss lifestyle and pharmacological interventions well. It would be helpful to briefly address the evidence level for each approach. For instance, metformin has shown mixed results in NAFLD, but pioglitazone has been more studied for NASH. Adding a concise comment on whether each approach has robust clinical trial data, preliminary animal data, or is primarily theoretical would give the reader more perspective.

Response 4: Thank you very much for your professional comments and suggestions on our work. We have carefully considered your suggestions and have revised the manuscript accordingly.I have taken lifestyle and pharmacologic interventions and made a brief description of the respective levels of evidence. For example, ( section 4.2.1.) supplementation with metformin has received mixed reviews for non-alcoholic liver, but pioglitazone has been studied more for NASH. And cite the EASL-EASD-EASO Clinical Guidelines for the Treatment of NAFLD and their specialized literature, along with a concise review of reliable clinical trial data, preliminary animal trial data, or primary theoretical data.We thank the reviewers again for this valuable suggestion.

Comments 5: :The mention of mitochondrial transplantation is intriguing. As this is a very new approach, you might add a short note on its current limitations (e.g., feasibility, safety in humans) to temper expectations.

Response 5 :We are extremely grateful for the professional remarks and recommendations you offered on our work. We have given careful thought to your suggestions and have made the appropriate amendments to our manuscript accordingly.I have briefly added, by searching scholarly repositories such as PubMed, ( section 4.2.4), the current limitations of mitochondrial transplantation, e.g., feasibility and safety for humans.We once again convey our deep gratitude to the reviewers for their significant and useful insights.

 Comments 6: Language (x) The English is fine and does not require any improvement.

Response 6: Thank you very much for your professional comments and suggestions on our work. We have carefully considered your suggestions and have revised the manuscript accordingly.For errors in the English expression, we have accepted the English editing service provided by MDPI because we are not native English speakers.We thank the reviewers again for this valuable suggestion.

Reviewer 2 Report

Comments and Suggestions for Authors

The authors have prepared narrative review about mitochondrial dysfunction in metabolic dysfunction-associated fatty liver disease. The topic of the manuscript is scientifically important. However, the authors have to carefully restructure the whole manuscript to prepare a version that would be acceptable for publication, by my opinion.

Some parts of text are repeated, some could be placed in different sections, and in some paragraphs relevant literature is not cited. 

Repeated text-lines 30-34, 45-47, 105-122, 348-355, 424-432, 440-454

Missed references-lines 233-238, 375-385, 455-472, 473-493, 498-516, 529-533, 562-564, 594-604, 606-624627-630, 666-679, 685-702

Please consider replacing of the text to appropriate sections- lines 279-288 Mitochondrial dysfunction in MAFLD, lines 592-594 Physical activity, weight loss

ATP production (section 2.2.1) could be described with more details, for example names of complexes could be added.

Mitochondrial-based MAFLD treatment could be carefully rewritten. The literature search strategy for this section could be added to provide evidence that all relevant studies were found and included as number of cited studies is rather small, and then all found references added and described.

The abbreviations should be checked as some abbreviations are induced several times and there is no description for some of them. 

Kind regards

Author Response

Comments 1: Some parts of text are repeated, some could be placed in different sections, and in some paragraphs relevant literature is not cited.

Repeated text-lines 30-34, 45-47, 105-122, 348-355, 424-432, 440-454

Missed references-lines 233-238, 375-385, 455-472, 473-493, 498-516, 529-533, 562-564, 594-604, 606-624,627-630, 666-679, 685-702

Response 1: We deeply appreciate your professional insights and recommendations regarding our work. Your valuable suggestions have been meticulously deliberated, and we have made the corresponding revisions to our manuscript. The specific modifications are as follows:

For the problem of repetitive content in the manuscript, we have critically revised the manuscript and deleted this part to ensure the conciseness of the manuscript.To address the problem of missing references, the correct professional references were added by searching PubMed and other academic databases. Lines 233-238 became lines 208-213, lines 375-385 became lines 361-371, lines 455-472 became lines 418-433, lines 473-493 became lines 434-455, lines 498-516 became lines 459-475, lines 529-533 became lines 487-491, lines 562-564 became lines 519-522, lines 594-604 became lines 558-568, lines 606-624 became lines 589-607, lines 627-630 became lines 624-643, lines 666-679 became lines 719-731, lines 685-702 became lines 746-792.The revised manuscript meets the journal's requirements for clarity and conciseness while maintaining scientific rigor. We thank the reviewers again for their valuable suggestions.

Comments 2: Please consider replacing of the text to appropriate sections- lines 279-288 Mitochondrial dysfunction in MAFLD, lines 592-594 Physical activity, weight loss

Response 2: Thank you very much for professional comments and suggestions for our work.We have carefully considered your suggestions, and have revised our manuscript accordingly.Regarding your point that the content needs to be replaced in the appropriate section ,we attach great importance to it and have carried out systematic verification and revision, the revision is as follows: (lines 279-288 Mitochondrial dysfunction in MASLD), we found that this part is not closely related to this subsection, so we have deleted this part of the content.

(Lines 592-594 Physical Activity, Weight Loss), we have already discussed this in detail in the (Section 4.1.2. Physical Activity and 4.1.3. Weight Loss) section, so we deleted this section. We are aware of the importance of academic standards and will follow up to further strengthen our ability to write professionally and avoid similar problems.Once more, we extend our sincere gratitude to the reviewers for their invaluable suggestions.

Comments 3: ATP production (section 2.2.1) could be described with more details, for example names of complexes could be added.

Response 3: We are very grateful for the reviewers' comments, which have important guiding significance for our manuscript. In response to your question about adding the name of the mitochondrial respiratory chain complex complex to make the description more detailed, we have made in-depth revisions and improvements. The specific revisions are as follows: in ATP production (Section 2.2.1), we have added an academic definition of complex I-complex V. We recognize the importance of accurate and complete academic statements in research papers, and we thank you again for your expert advice, which has greatly improved the academic rigor of this paper.

 Comments 4: Mitochondrial-based MAFLD treatment could be carefully rewritten. The literature search strategy for this section could be added to provide evidence that all relevant studies were found and included as number of cited studies is rather small, and then all found references added and described.

Response 4: Thank you very much for professional comments and suggestions for our work.We have carefully considered your suggestions, and have revised our manuscript accordingly.You pointed out that the chapter on mitochondrial dysfunction in the treatment of MAFLD was insufficiently cited in the literature, which pointed out the direction of focus for the improvement of this paper. We attach great importance to this suggestion, and have systematically supplemented and deepened the relevant content, with the following specific improvement measures: in the section (Section 4. Mitochondria-based treatment of MASLD), we have carefully revised the content of the section and correctly cited the references. We thank the reviewers again for this valuable suggestion.

Comments 5:The abbreviations should be checked as some abbreviations are induced several times and there is no description for some of them.

Response 5: We are extremely grateful for the professional remarks and recommendations you offered on our work. We have given careful thought to your suggestions and have made the appropriate amendments to our manuscript.. We have systematically checked and optimized the acronym system of the whole paper, and the specific improvements are as follows: the missing acronyms have been added to the position of the acronym list of the manuscript, and a standardized description has been made in the abstract for the search literature strategy. We express our renewed appreciation to the reviewers for their precious suggestions.

Comments 6: Language (x) The English is fine and does not require any improvement.

Response 6: Thank you very much for your professional comments and suggestions on our work. We have carefully considered your suggestions and have revised the manuscript accordingly.For errors in the English expression, we have accepted the English editing service provided by MDPI because we are not native English speakers. We once again convey our deep gratitude to the reviewers for their significant and useful insights.

Round 2

Reviewer 2 Report

Comments and Suggestions for Authors

The authors have answered to all my requests, and manuscript could be accepted in present form, by my opinion.

Author Response

Comments 1: The authors have answered to all my requests, and manuscript could be accepted in present form, by my opinion.

Response 1: We sincerely appreciate the time and effort you have put into evaluating this manuscript and are honored to learn that you consider the revised manuscript to have met the requirements. Each of your suggestions during this revision process has provided important direction for improving the quality of the manuscript, allowing me to continually optimize the logic of the content and the academic presentation. As researchers dedicated to the advancement of molecular endocrinology and metabolism, we highly value this collaborative process of improving academic work through peer review. Thank you again for your support and guidance.

Comments 2: Language (x) The English is fine and does not require any improvement.

Response 2: Thank you very much for professional comments and suggestions for our work.I have entrusted the professional language editing team appointed by this journal to carry out in-depth touch-up processing of the manuscript. This touch-up strictly follows the journal's submission specifications and language style requirements, and systematically optimizes the manuscript in terms of the accuracy of academic terminology, compliance with grammatical structure, fluency of expression, and other dimensions. We thank the reviewers again for this valuable suggestion.
